# MIRAGE: MODELLING INTERPRETABLE MULTIVARIATE TIME SERIES FORECASTS WITH ACTIONABLE GROUND EXPLANATIONS

## ABSTRACT

Multi-variate Time Series (MTS) forecasting has made large strides (with very negligible errors) through recent advancements in neural networks, e.g., Transformers. However, in critical situations like predicting a death in an ICU or sudden gaming overindulgence affecting ones mental well-being; an accurate prediction (forecast) without a contributing evidence (explanation) is irrelevant. Hence, it becomes important that the forecasts are *Interpretable* - intermediate representation of the trajectory is comprehensible; as well as *Explainable* via input features, allowing prevention of the incident; e.g., controlling BP to avoid death, or nudging players to take breaks to prevent overplay. We *introduce* a *novel* deep neural network, MIRAGE, which addresses the inter-dependent challenges associated with three exclusive objectives - 1) forecasting accuracy; 2) smooth comprehensible trajectory and 3) explanations via input features on the high multi-dimensional data forecasts — which has yet not been addressed together in one single solution. MIRAGE, our novel deep neural network: (i) achieves over 85% improvement on the MSE of the forecasts in comparison to the most relevant interpretable SOM-VAE based SOTA networks; and (ii) unravels and attributes the progression of the multi-variate time series to a specific actionable inputs feature(s), establishing itself to be a first of its kind.

## 1 BACKGROUND AND MOTIVATION

**Background:** MTS forecasting has made large strides (with very negligible errors) through recent advancements in neural networks, e.g., Transformers. However, in critical situations like predicting a death in an ICU or sudden gaming overindulgence affecting ones mental well-being; an accurate forecast without a contributing evidence (explanation) is irrelevant. Hence, it becomes important that the forecasts are *Interpretable* - trajectory in the representation space is comprehensible; as well as *Explainable* - attentive features in the input space, allowing prevention of the incident. *Interpretability* and *Explainability*, though often used interchangeably, are different in the sense that the former answers the "how" and the latter answers the "why" about the prediction. Interpretability refers to the process of associating the complex representation space learnt by a neural network to a simpler and comprehensible pattern which a domain expert can validate. Explainability on the other hand, is the process of associating the decision making of the model to a set of few input features which the model was attentive to and can be acted upon by the end user of the model. Exclusivity between prediction accuracy, explainability and interpretability has led to most of the state of the art (SOTA) models target a single objective, prediction accuracy being the most common.

Work described in SOM-VAE Fortuin et al. (2019) and T-DPSOM Manduchi et al. (2021) to the best of our knowledge, have been the only SOTA networks focusing on learning a discrete interpretable representations on time series data but with negligible attention on forecasting. Authors introduce a novel concept of "*smoothness*" over temporal trajectory as an important requirement for interpretable time series models. They argue that the representations learnt over time series data make misleading i.i.d. assumptions making interpretability difficult. To enforce this temporal smoothness, they propose to reduce the high dimensional representation space into a two-dimensional grid enforcing Self Organised Map (SOM) Kohonen (1990) like topological neighbourhood. Temporal smoothness is then facilitated by encouraging a higher probabilistic likelihood that temporally

adjacent data points belong to the same or immediately next SOM centroid. This is done via a probabilistic transition model in Fortuin et al. (2019) and a LSTM based model in Manduchi et al. (2021).

**Motivation:** Though we agree that i.i.d. assumptions hinder the interpretability of the model, it is unlikely that every real-world dataset exhibits a linear temporal progression. Authors primarily validated interpretability on synthetic time series of linear interpolations on MNIST handwritten digits LeCun et al. (1998), Fashion-MNIST Xiao et al. (2017). The other real world dataset used for validation (medical data from the eICU Collaborative Research Database ( Goldberger et al. (2000); Pollard et al. (2009)) containing vital sign time series measurements of intensive care unit (ICU) patients), as we analysed was much linear or non-random.

**Test of Randomness:** In this paper we primarily deal with players game play time series data on a gaming platform and we argue that this data is highly random and non-linear. In order to assess the degree of randomness/non-linearity in our

|  | eICU Dataset | Players Dataset |
|---|---|---|
| Runs Test (p-value) | $0.0197 \pm 0.0071$ | **$0.2877$** $\pm 0.1214$ |
| Auto-correlation Test | $0.9554 \pm 0.0307$ | **$0.2723$** $\pm 0.0856$ |

Table 1: Statistical tests for Randomness in time-series data.

data w.r.t the eICU dataset previously used, we performed Runs Test ( Bujang & Sapri (2018)) and 1-lag auto-correlation test ( Huitema & Laraway (2006); pea (2008)) respectively. We randomly sampled 1000 users from both the datasets and took average across features. Table 1 shows that the p-value of players data is greater than 0.05 indicating not strong enough evidence to reject the null hypothesis of a random process, contrary to eICU data. Table 1 also shows the 1-lag auto-correlation results with eICU dataset features showing high auto-correlation co-efficient compared to the players data.

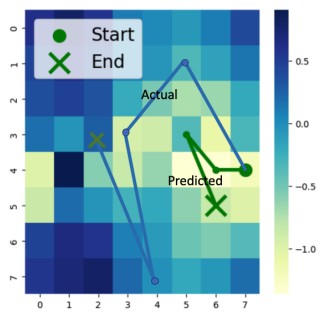

Figure 1: DPSOM mispredicted temporally smooth trajectory forecast for a non-overindulgent player vs. the actual

**Source of Non-linearity:** Though gaming patterns are quite outcome based (win/loss), the psychological imprints on the player from their previous game play could generate very diverse game behaviours in immediate next play ( Talwadker et al. (2022)) - player with previous losses could show assured and confident game play style while somebody with high winnings may actually show unnecessary aggression, and vice versa. Hence the factors affecting the future predictions (co-variates) are not completely observed, measurable, or generalizable. **2. Generic Model for Multiple Process Modeling:** The time series in the traditional datasets (WTH ncei, ECL Trindade (2015), ETTh, ETTm Zhou et al. (2021a) etc.) have been predominantly generated by a single process - e.g electricity demand of a city. In our case the data comes from many players - synonymous to many processes and its generalization is a new challenge.

Figure 1 illustrates a 2-D intermediate representation space (to be discussed in detail later), generated using the SOTA Manduchi et al. (2021) model on our player data. The colour of each position (cell) in the grid indicates the mean of the overindulgence score of the data points mapped to it (darker shade indicating higher gaming addiction). The temporal trajectory is generated by the model, which is seen to remain in the lighter zone until the end of forecast. The actual (expected) trajectory (mapped using the actual data, in dotted line) on the contrary shows progression into a dark zone, indicating gaming addiction. This misprediction shows inability of the SOTA methods to model a non-linear trajectory and preference towards a smooth trajectory for interpretability. The actual trajectory, however, shows long jumps and is quite non-smooth for interpretability and acts as a motivation towards building MIRAGE.

Our proposed deep neural network, *MIRAGE* needs to preserve the smoothness (for interpretability), without compromising the prediction accuracy while dealing with such non-linear datasets. MIRAGE, towards addressing these challenges makes the following **contributions:**

- Discrete Markov Model (DMM): We contend that the nature (linear/random) of the immediate next data point can be attributed to the present state of the process, e.g. players' psychological imprint from the previous experience. MIRAGE trains a DMM to infer the state associated with the time series and uses it guide the next prediction.

- Re-parameterization of the latent space: Learnt representation space by the Variational AutoEncoder (VAE) ( Kingma & Welling (2019)) in SOM-VAE is further diversified by replacing it with a conditional VAE (cVAE) Sohn et al. (2015).

- Joint optimization of the temporal smoothness and forecasting accuracy via learning a *damping factor* which provides a "leeway" to perform a non-smooth, far-off jump. The corresponding non-smooth trajectory is then explained via the discrete state predicted by the DMM .

- Rigorous quantitative validation as well as case studies, as applicable, on variety of industry standard benchmarks and datasets for the three exclusive target objectives in one single instance of model training.

Our code base and the relevant datasets are available at MIRAGE_Git (2023)

## 2 RELATED WORK

**Multivariate Time Series Forecasting:** MTS forecasting models are roughly divided into statistical (Vector auto-regressive (VAR) model Kilian & Lütkepohl (2017), Vector auto-regressive moving average (VARMA)) and neural networks. Statistical models assume a linear cross-dimension and cross-time dependency. With the development of deep learning, many neural models often empirically show better performance than statistical ones. LSTnet Lai et al. (2017) and MTGNN Wu et al. (2020) use CNN and graph neural networks respectively for cross-dimension and RNN for cross-time dependencies. RNN's, however have empirically shown to have difficulty in modeling long-term dependency. **Transformers:** Transformers Vaswani et al. (2017) were introduced in natural language processing (NLP). Recently, many Transformer-based models have been proposed for MTS forecasting, showing great results. Informer Zhou et al. (2021b) proposes ProbSparse self-attention which achieves $\mathcal{O}(L \log L)$ complexity. Autoformer Wu et al. (2021) introduces an Auto-Correlation mechanism to Transformer. Pyraformer Liu et al. (2021) introduces a pyramidal attention module that summarizes features at different resolutions and models the temporal dependencies of different ranges. FEDformer Zhou et al. (2022) develops a frequency enhanced Transformer for the time series that have a sparse representation in the frequency domain. Following up with the improvements Zhang & Yan (2023) further focused on the cross-dimension dependency and Shabani et al. (2023) introduced iterative scale refinement and cross-scale normalization on any of the above transformer based architectures which further pushed the MSE baselines to an extra-ordinary level. Though quite impressive, these works do not offer any explanations over the prediction space, except Lim et al. (2021) which however, make strict assumptions on presence of seasonality in time series data. **Interpretability:** In recent years, interpretability has increasingly been combined with generative modeling through the advent of generative adversarial networks (GANs) Goodfellow & al. (2014); Mirza & Osindero (2014) and variational autoencoders (VAEs) Kingma & Welling (2019). However, the representations learned by these models are often considered cryptic and do not offer the necessary interpretability Chen et al. (2016). A lot of work has been done to improve them in this regard Higgins et al. (2017); Wu et al. (2022). Nonetheless, these works have focused entirely on continuous representations, while discrete ones are still being under-explored. Discrete representations are known to reduce the problem of "posterior collapse" van den Oord et al. (2017); Razavi et al. (2019), situations in which latents are ignored when they are paired with a powerful auto-regressive decoder typically observed in the VAE framework. Fortuin et al. (2019); Manduchi et al. (2021) introduce an impressive method of achieving discrete space interpretability for high-dimensional time series, by working on SOM like clustering in a low-dimensional latent space. Major drawback being, assumption of temporal smoothness in the time series data and lacking a capability to forecast into the future. **Explainability:** Explainability allows one to comprehend a particular prediction w.r.t the input features. SHAP Lundberg & Lee (2017) assigns each feature an importance value for a particular prediction. Traditionally, SHAP and LIME Ribeiro et al. (2016) have been used to identify feature weights for a multi-class classification or a regression based models. It still remains to be a challenge in the case of an unsupervised model, where there is no such response variable/class to hook to. Secondly, when we talk about explanations over a time evolving data points, how do we narrow down to the the set of core features as each time step would provide its own set of important features? **Deep Markov Models:** The Deep Markov Model in Meng et al. (2019) introduces continuous latent variables, non-linear transition networks, and non-linear emission networks. However, it presents an unsupervised framework which cannot be guided via

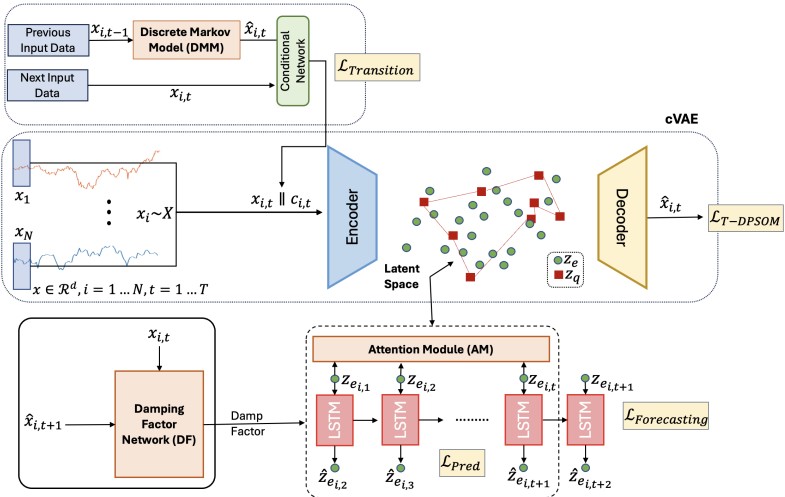

Figure 2: MIRAGE Proposed Architecture

already known facts. Ozyurt et al. (2021) relies on posterior approximation of the latent Markov state, because of which, its application in a forecasting task is precluded. Moreover, the observed improvements in *AUROC* and *AUPRC* are primarily attributed to smooth time varying patient physiology.

## 3 MIRAGE NETWORK

SOM-VAE first proposed model for learning interpretable representations over the time series data. T-DPSOM extended over SOM-VAE, by introducing soft probabilistic clustering and LSTM based temporal mappings, showcasing much better results. MIRAGE is built over T-DPSOM and we refer to it as our prior art. Our proposed architecture is presented in Figure 2. Training workflow is available in Appendix B

As shown in the Figure 2, MIRAGE consists of a Discrete Markov Model (DMM), a conditional VAE based Encoder-Decoder and a LSTM network. In MIRAGE, along with the introduction of DMM, the LSTM network is also modified with attention module (AM) and a damping factor network (DF) which is used to dampen the penalty due to non-smooth (distant) temporal jumps. A forecast fine tuning (FFT) step is introduced to further reduce the MSE on forecasts. Figure 3 shows a simplistic block diagram of MIRAGE.

### 3.1 INTERPRETABLE FORECASTING IN MIRAGE

MIRAGE realigns the discrete representation space associated with the SOM clusters such that, temporally adjacent non-linear and topologically adjacent linear observations are competitively mapped. The objective is to exploit the forecasting performance as well as, to largely preserve interpretability. This could be possible if - 1) MIRAGE is able to understand, as a part of its learning process when the upcoming observations are non-linear; 2) it is able to train these distinct feature distributions (assuming the two types of data points come from different distributions) as a part of a single embedding space for interpretability. *Note:* SOM space is referred to as embeddings (in the prior art) as they follow a topological structure.

#### 3.1.1 DISCRETE MARKOV MODEL (DMM):

MIRAGE incorporates "Latent State Prediction" via a Discrete Markov Model (DMM) which is collaboratively trained to predict different states (conditions) which probabilistically generate the two types of (linear/non-linear) data points. DMM is pre-trained prior to training the cVAE based latent space and the LSTM based temporal transition model. We hypothesize that each state is associated with certain observations (coming from a specific distribution) and change in the nature

of observations could be related to a change in the state. We refer to a state as "condition" since it associates directly with the term 'condition' in a conditional VAE. Since in a Markov Model the next state only depends on the prior one, using the past observations the state of the immediate future observation can be learned. To facilitate its learning MIRAGE adopts a feed forward network with two types of metrics for each sample $i$: 1) error in prediction of the next observation given the immediate past observation as: $\mathcal{L}_{MSE} = \sum_{i=1}^{N} \sum_{t=1}^{T} \frac{1}{d} \sum_{j=1}^{d} \parallel x_{i,t,j} - \hat{x}_{i,t,j} \parallel_2$ and 2) a probability distribution of both the predicted and the actual observation via a dense layer with, number of conditions, $s$ as a hyper-parameter using a Categorical Cross Entropy loss; $\mathcal{L}_{Conditional} = -\sum_{i=1}^{N} \sum_{t=1}^{T} \frac{1}{s} \mathcal{P}_{\text{model}}(x_{i,t}) \log(\mathcal{P}_{\text{model}}(\hat{x}_{i,t}))$, where $\mathcal{P}_{\text{model}}()$ is an additional dense layer in the DMM. $X$ is an input set with $N$ time series for $x \in X$. For details on exact steps for training the DMM, refer to AppendixB. The total transition loss of the DMM is given by:

$$\mathcal{L}_{Transition} = \mathcal{L}_{MSE} + \mathcal{L}_{Conditional}$$

### 3.1.2 LATENT SPACE AND CVAE:

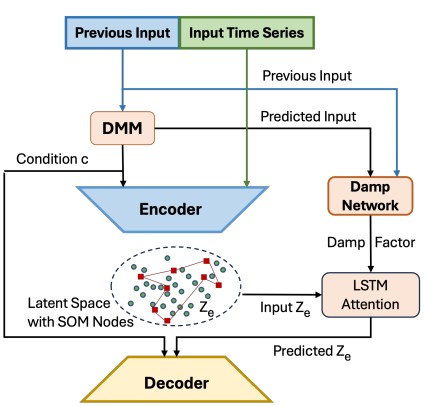

Figure 3: Block diagram of MIRAGE

MIRAGE uses a conditional VAE (cVAE) which leverages the conditions predicted by the DMM. This ensures that the data from various different Markov conditions is mapped to the same latent space, over which SOM grid could be enforced as done in the prior art. cVAE maps each input $x_{i,t} \parallel c_{i,t}$, where $x_i$ represents a single time series and $c_i$ refers to the discrete conditions from the DMM, to a latent encoding $z_e i, t \in \mathcal{R}^m$ (usually $m < d$ , $m$ - latent dimension and $d$ - data dimension) by computing $z_e = f_\theta(\text{x})$. Similarly, it also learns the probability distribution of the reconstructed output given a sampled latent embedding $p_\theta(x | z_e, c)$, where $(\mu_\phi, \sigma_\phi) = f_\phi(x, c)$ and $(\mu_\theta, \sigma_\theta) = f_\theta(z_e, c)$; $f_\phi$ , $f_\theta$ being the encoder and decoder networks respectively. SOM training step maps each encoding $z_e$ to a embedding $z_q \in \mathcal{R}^m$. Total number of embeddings is a hyper parameter and is derived empirically. In MIRAGE, the SOM space is set to a $8 \times 8$, 2-D space, which results in 64 discrete SOM embeddings. It was fixed empirically with least MSE/MAE and best NMI metrics. We use the same ELBO loss as in T-DPSOM for cVAE, however while passing the $c$ condition to the decoder for the reconstruction we apply a gradient stopping operator on the DMM predictions. This is being done to prevent SOM or cVAE network losses from influencing the conditions generated by the Markov model. We borrow the loss metric - $\mathcal{L}_{\text{T-DPSOM}}$, as is from the T-DPSOM paper and is defined as:

$$\mathcal{L}_{\text{T-DPSOM}} = \beta \mathcal{L}_{\text{SOM}} + \gamma \mathcal{L}_{\text{Commit}} + \theta \mathcal{L}_{\text{Reconstruction}} + \kappa \mathcal{L}_{\text{Smoothness}}$$

### 3.1.3 COMPETITIVE TRAINING OF SMOOTHNESS VS. ACCURACY:

T-DPSOM, in order to enforce smoothness of transitions, adds a smoothness term to maximize the similarity between latent embeddings of adjacent time steps to discourage large jumps in the latent space. Further, a long short-term memory network (LSTM) Hochreiter & Schmidhuber (1997) is added, which predicts the probability distribution over the next latent embedding, $p(z_{i,t+1} \mid z_{i,t})$ using Log-likelihood loss between actual and predicted $z$ distributions. To aid accurate forecasting while preserving a smooth trajectory, MIRAGE introduces - 1) a damping factor which provides leeway for larger jumps in the latent space; 2) a self-attention layer over the temporal latent embeddings to boost the accuracy of the temporal predictions and 3) forecasting fine tuning via MSE loss on the actual future predictions.

*Damping Factor (DF module):* Damping Factor is added to the LSTM prediction loss as:

$$\mathcal{L}_{\text{Pred}} = -\sum_{i=1}^{N} \sum_{t=1}^{T-1} \log p_\omega(z_{i,t+1} | z_{i,t}) \, \mathcal{D}_{\text{model}}(x_{i,t}, \hat{x}_{i,t+1})$$

$p_\omega$ refers to the probability distribution over the next latent embedding (parameterized by $z$). The damp factor network (DF), $\mathcal{D}_{\text{model}}()$ is tasked with a new objective to learn a damping factor between older input $x_{i,t}$ and predicted next input $\hat{x}_{i,t+1}$ for $t \in \{1 \ldots, T-1\}$. To train the damping factor network, the actual previous observation and the next observation predicted by the DMM are concatenated and passed through dense and batch normalization layers before being passed through a Sigmoid layer which generates a scaled output between $[0, 1]$. In a case where the transition is not smooth (to a far away SOM grid position) the low likelihood generated by the prediction loss when passed through the logarithmic function would generate a high loss value (penalty). In the traditional T-DPSOM based LSTM training such a transition would get discouraged. To compensate in favour of the forecasting loss, LSTM is offered an option to dampen this effect by simultaneously training the damping factor network. The damping loss is backpropagated to the SOM training layer, which also realigns its embedding (SOM cluster) space to preserve interpretability

*Forecasting Fine-Tuning (FFT) with Self Attention:* MIRAGE learns a self-attention module (AM) over all the previous latent embeddings while predicting embeddings for the next time step.

*Forecasting:* LSTM network generates next latent encoding, $P_\omega(z_{i,t+1}|z_{i,t})$ which is then passed to the decoder to generate $\hat{x}_{i,t+1}$. Forecasting fine tuning ($\mathcal{L}_{\text{Forecasting}} = ||x_{i,t+1} - \hat{x}_{i,t+1}||_2$) is done post the LSTM prediction step. The final loss function in MIRAGE is given as:

$$\mathcal{L}_{\text{MIRAGE}} = \tau \mathcal{L}_{\text{Transition}} + \mathcal{L}_{\text{T-DPSOM}} + \eta \mathcal{L}_{\text{Pred}} + \mathcal{L}_{\text{Forecasting}}$$

The various weight parameters and their values are discussed in the Appendix and also are available at the MIRAGE GitHub MIRAGE_Git (2023). Hyper parameters were trained using Optuna Akiba et al. (2019) and are mentioned in Appendix A

### 3.2 ENABLING FEATURE BASED EXPLAINABILITY IN MIRAGE:

**Attention Time Step:** MIRAGE leverages attention weights of the LSTM network to identify which present time step is the predicted future time step attentive to for proactive intervention.

**SHAP based Correlated features and more:** The attention mechanism helps identify the time of intervention, however, the exact modality of the intervention can only come via explanations. With regard to the challenges of identifying the regressor and merging explanations across multiple time steps (discussed in the Section 2), MIRAGE provides the following options- the first problem is resolved by letting the SHAP Lundberg & Lee (2017) model regress its Shapley values on the SOM cluster identities as the y-label and the actual game play features as the corresponding x-values. For the temporally changing features over the SOM map; MIRAGE compares and prioritizes top Shapley values at each of the attentive time steps and defines a function which measures the extent of acceleration or deceleration amongst them over time, to point out one or few unique features which would continue to change and hence should be intervened.

**Markov Condition that initiated the Trigger:** Condition from the DMM model also provides additional explainability. We found that certain problematic trajectories are inherently a part of specific Markov conditions which could also act a proactive warning.

## 4 EVALUATION

**Datasets:** *Player Dataset:* consists of 19 dimensional time series data for over 5 months of weekly time steps, for about 50K players on a gaming platform. train/val/test sets are normalized into the range of $[0, 1]$ to handle varying and sparse data dimension scales and were split in the ratio of 0.7:0.1:0.2 as standard practice. *Real-World MTS Datasets:* evaluated in Zhang & Yan (2023): 1) ETTh1 (Electricity Transformer Temperature-hourly) 2) ETTm1 (Electricity Transformer Temperature-minutely), 3) WTH (Weather), 4) ECL (Electricity Consuming Load) with train/val/test split ratio of 0.7:0.1:0.2 as in Zhang & Yan (2023). 3) *Real-World Medical Data:* from the eICU Collaborative Research Database Goldberger et al. (2000); Pollard et al. (2009).

### 4.1 T-DPSOM VS MIRAGE:

*Forecasting Accuracy:* T-DPSOM (and SOM-VAE) fits the entire time series into a representation space during the training phase. We train it on a partial time series data and predict the latter time steps using its LSTM's prediction likelihood criteria. Figure 4 shows 3 distinct trajectories for a

player showing healthy game play behaviour throughout time of observation, 13 weeks. The SOM clusters are coloured by degree of players' gaming overindulgence, given by a players' risk score Chakrabarty et al. (2021) (dark clusters indicating higher risk). Figure 4a shows a 2-D SOM based trajectory of the player as fitted by T-DPSOM when all 13 time steps were learnt by it. Figure 4b, which is only trained on the first 7 time steps, indeed predicts that the player will move to higher levels of indulgence, which is incorrect. Primary assumption of temporal linearity and optimizing on neighbouring transitions in the LSTM training are the two major reasons for this misprediction. MIRAGE , Figure 4c when trained for the first 7 time steps is able to forecast a continuance into a healthy indulgence.

*Explainable Health Risk Predictions:* In the Figure 5a, MIRAGE correctly predicts death in the ICU with the patient's forecasted trajectory (dotted lines) moving from lighter to darker zones. Cluster colours indicate degree of patient abnormality measured by APACHE-24 scores W A & al. (1985) (more dark means "less healthy"). MIRAGE leverages the attention module (AM) and SHAP modules to provide a supporting evidence on what led to the prediction. It indicates a attention time step (shown as the black dotted vertical line) and points to the Hct vital sign as the attentive feature. Comparing the plot of Hct along with few other vitals shown to be unimportant by MIRAGE, Hct shows a rise while others showing no change. Figure 5b depicts a case where the patient survived with forecasts remaining in lighter zones. MIRAGE points that at the attentive time point patient's bilirubin levels dropped. For patient health data, due to lack of any ground truth we could only present a representative example. We believe that MIRAGE will significantly elevate the state of predictive and preventive health care with such proactive causal explanations.

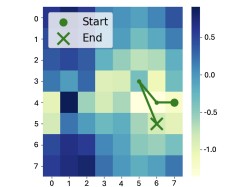 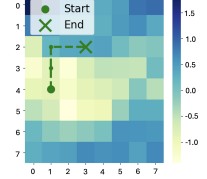 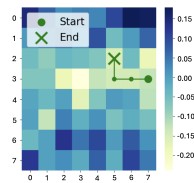

(a) Player trajectory with T-DPSOM trained on 13 weeks

(b) Player trajectory with T-DPSOM trained on 7 weeks

(c) Player trajectory with MI-RAGE trained on 7 weeks

Figure 4: Player Trajectory Forecast (last 6 instances): player showing healthy gaming in the future; T-DPSOM when partially trained, predicts overindulgence due to temporal neighbourhood constraint

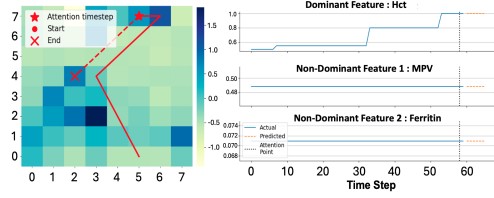 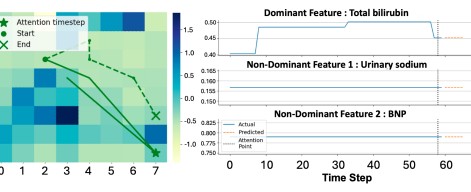

(a) MIRAGE: Patient Died in ICU

(b) MIRAGE: Patient Survived

Figure 5: Accurate Interpretable Predictions with Explanations for ICU Patients. Dominant feature is the one over which attention was drawn by the model. Non-dominant features show no change in the trend as the trajectories change. The features being human vital signs Goldberger et al. (2000).

*Preventing Gaming Overindulgence with proactive Nudges:* Figure 6 shows a trajectory for a player who was identified to be overindulgent (going into the darker zone). Dotted lines indicating the forecast. In the Figure 6a, we show two attention points, The first attention point is for the first forecasted time step and the second is for the final forecasted step provided by MIRAGE. The Explanations indicates the causal features. More on game features is discussed in Appendix B. In Figure 6b, MIRAGE attributes player's overindulgence to high add cash amount (acf_amount) while the final culmination into the darkest zone is attributed to increase in add cash failures (acf), indicating player's mental desperation.

Using MIRAGE, one can proactively nudge the player by limiting player's add cash limit on the gaming platform. Figures 6c and 6d show importance map (darker indicating high importance) of the two gaming features - acf and acf_amount. We see that the intensity of the two features are high at the regions corresponding to the respective attention points. Figure 6a shows the conditions (player psychology states) predicted at each point by the DMM. All of the transitions into darker zones are indicated by condition 2, which could indicate an "aggressive" game play state of the player. This further aids interpretability when trajectory is non-smooth.

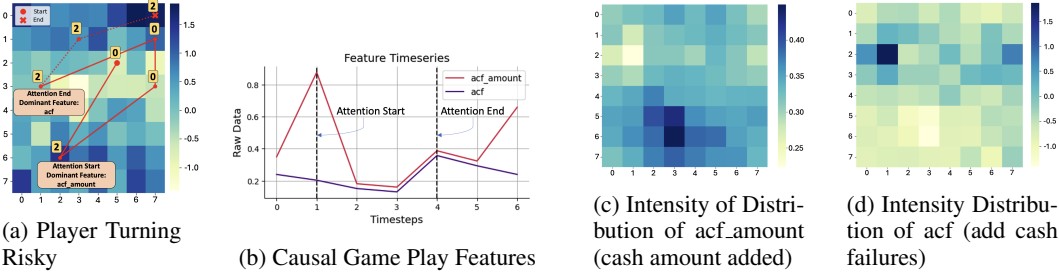

(a) Player Turning Risky

(b) Causal Game Play Features

(c) Intensity of Distribution of acf_amount (cash amount added)

(d) Intensity Distribution of acf (add cash failures)

Figure 6: Interpretable Forecasting and Explainability on the Gaming Player's Trajectory.

**Overall Performance Comparison:** Table 2 compares performance of T-DPSOM with MIRAGE with, T-DPSOM trained for 7 time steps of player data and 66 time steps of patient data and predicted using its LSTM network for the rest. MIRAGE clearly outperforms T-DPSOM with over 85% improvements in the MSE values. Overindulgence being a scarce phenomenon Chakrabarty et al. (2021), its forecast is quite challenging. Note that, unlike in patient

| Dataset | T-DPSOM | MIRAGE |
|---------|---------|--------|
| eICU data | 0.1 | **0.012** |
| Player data | 0.5 | **0.039** |

Table 2: MSE on Forecasts : T-DPSOM vs. MIRAGE. MIRAGE clearly outperforms

data where a single step into a darker zone indicates fatality, player can be in and out of the over-indulgence zone indicating self-moderation. MIRAGE achieves a recall of 40% with a precision of over 60% in predicting sustained overindulgence vs. healthy game play. We define sustained overindulgence as the player continuing to being into a darker zone for two or more consecutive time steps. T-DPSOM largely under-performs with recall of 15% at a precision of under 20%

**Quantifying Interpretability:** With player data, neighbouring transitions could be non-smooth, hampering interpretability. MIRAGE leverages condition switch as an explanation to justify it. To validate if correlation between a long jump and condition switch is dependable, we created new time series by calculating shortest distance between the clusters assigned to the two successive time steps and correlated with the change in conditions predicted at each point. We found the Pearson's correlation pea (2008) to be primarily high and positive in most cases (**0.798** for example in Figure 6). This exercise proved that the transition to farther nodes are mostly associated with a change in underlying Markov condition.

**Ablation Study:** In our approach, there are four novel components: Discrete Markov Model - **DMM**, Attention Module - **AM**, Damping factor - **DF** and Forecasting Fine-tuning - **FFT**. In the Table 4, MIRAGE refers to **DMM+ AM+ DF+ FFT** i.e. no ablation. We've performed ablation over these components to study their effect on the MSE and temporal trajectories using player data.

Table4 shows the MSE results for each experiment (detailed graphs in Appendix B) DMM helps to capture the parameter distribution well, especially for non-linear player data and in its absence MSE could be high due lack of guiding loss function for non-linear data. Without AM, the MSE starts well, as DMM is mapping the space well, however the rise in MSE during prediction fine tuning is significant due to lack of long term pattern purview. Without DF, the model is not able to predict farther nodes leading to higher MSE. MIRAGE outperforms all the ablations.

## 4.2 REAL WORLD MTS DATASETS:

Recent, transformer based MTS forecasting networks show negligible prediction errors. However, most fail to provide no or very little insight on why such a prediction has been made.

In the Table 3 we show comparative results using identical data scaling on a few real world datasets. MIRAGE does better than LSTNet Lai et al. (2017) (which is one of the best industry standard

| Dataset | Prediction Length | LSTnet | | MIRAGE | | CrossFormer | |
|---|---|---|---|---|---|---|---|
| | | MSE | MAE | MSE | MAE | MSE | MAE |
| ETTh1 | 24 | 1.293 | 0.901 | 0.483 | 0.634 | 0.305 | 0.367 |
| | 48 | 1.456 | 0.960 | 0.580 | 0.790 | 0.352 | 0.394 |
| | 168 | 1.997 | 1.214 | 0.826 | 0.954 | 0.410 | 0.441 |
| ETTm1 | 24 | 1.968 | 1.170 | 0.397 | 0.478 | 0.211 | 0.293 |
| | 48 | 1.999 | 1.215 | 0.551 | 0.577 | 0.300 | 0.352 |
| | 168 | 2.762 | 1.542 | 0.742 | 0.878 | 0.320 | 0.373 |
| ECL | 24 | 0.369 | 0.445 | 0.717 | 0.815 | 0.156 | 0.255 |
| | 48 | 0.394 | 0.476 | 0.948 | 0.990 | 0.231 | 0.309 |
| | 168 | 0.419 | 0.477 | 1.281 | 1.202 | 0.323 | 0.369 |
| WTH | 24 | 0.615 | 0.545 | 1.083 | 0.968 | 0.294 | 0.343 |
| | 48 | 0.660 | 0.589 | 1.241 | 1.037 | 0.370 | 0.411 |
| | 168 | 0.748 | 0.647 | 1.695 | 1.236 | 0.473 | 0.494 |

Table 3: Evaluation: Crossformer Zhang & Yan (2023) outperforms MIRAGE as it's primarily tuned for prediction. MIRAGE beats LSTNet which is one of the best benchmarks besides the Transformer based architectures.

| Training Steps | No DMM | Without AM | Without DF | MIRAGE |
|---|---|---|---|---|
| MSE after Full Training | 0.1118 | 0.0596 | 0.0680 | **0.0516** |
| MSE after Prediction Fine tuning | 0.1205 | 0.0838 | 0.0680 | **0.0620** |
| MSE after Forecasting Fine tuning | 0.1472 | 0.0803 | 0.0715 | **0.039** |

Table 4: MSE results on component ablation using player data.

benchmarks after the Transformer based models) on ETTh1 and ETTm1 datasets. LSTNet uses a CNN to extract cross-dimension and LSTM for cross-time dependencies for forecasting.

**Interpretability and Explainations for the First time:** The Figure 7 shows two interpretable trajectories. In both of these trajectories, MIRAGE correctly predicts (dotted lines) the target variable to be either going from high to low or vice versa (heatmaps correspond to the intensity of the target variable OT). Darker zones indicate higher values of the variable. The adjacent time series plots indicates attention points and the respective features being important for the trajectory of OT. There is no prior available so as to how these various features influence each other during forecasting. As an example we observe that target variable OT in the Figure 7a in the forecasted time period continues at a constant low value with some intermittent small peaks. MIRAGE largely associates this predictability to a feature HUFL, rather then the past values of OT itself, indicating **cross dimension dependency**. This looks statistically convincing as the past values of OT have been quite high and do not quite justify the drop while the HUFL values show a rise post a heavy drop in the past. Least dominant feature LULL , pointed by MIRAGE too shows a dip at the attention point. MIRAGE provides a feature importance map for each cluster in the grid (see Appendix), having known the attention point in the trajectory one could easily trace the attentive causal feature.

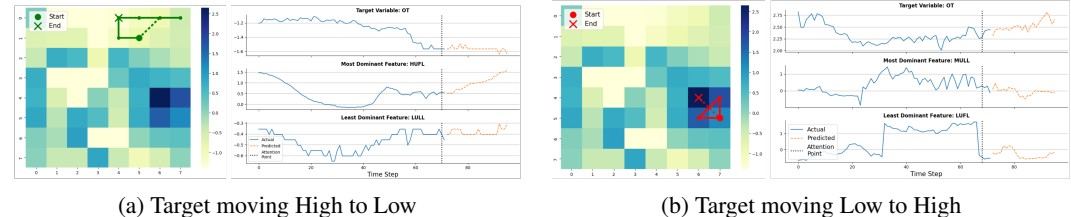

(a) Target moving High to Low          (b) Target moving Low to High

Figure 7: ETTm1:Solid line denotes input and the dotted line denotes the projected time series

## 5 CONCLUSION

We have proposed MIRAGE, a novel deep neural network for interpretable forecasts and explanations over the multi-dimensional temporal feature space. We specifically target forecasts on very commonly encountered non-smooth, chaotic time series datasets and demonstrate very unique findings. There are two systematic exploration possibilities that one could extend forward from this work. Firstly, one could revisit the logic for reconstruction of a sample from the VAE based network to further improve the forecasts. Secondly, one could explore a different latent space structure (apart from SOM), for example a 2D grid, by learning the data relationships as a graph and enforcing those learnings on the latent space.

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

## A   APPENDIX

## B   DETAILS OF EXPERIMENTS

### B.1   TRAINING WORKFLOW

---

**Algorithm 1** MIRAGE Training Workflow

---

**Require:** $D = \{p_1, \ldots, p_k, x_1, \ldots, x_d | \forall p_i \text{ and } x_j \in \mathbb{R}\}$, labels $L = \{p_1, \ldots, p_k, y_1, \ldots, y_l | \forall p_i \in \mathbb{R} \text{ and } y_j \in \mathbb{I}\}$, $k$ is number of clusters formed using $C$ steps of time series data, $b$ is total number of features at time $t$, neighbourhood function $N(.)$

**1. DMM Pretraining**

**for** each pretrain-epoch **do**

   Train DMM with proportional vector $\{ p_1, \ldots, p_k \}$ from $D$ as input and $\{ p_1, \ldots, p_k \}$ from $L$ as output.

   Optimize DMM network to generate corresponding conditions $c$.

**end for**

**2. cVAE Pretraining**

**for** each pretrain-epoch **do**

   Encode the input data $\{ x_1, \ldots, x_d, \}$ along with $c$ condition through encoder into $m$ dimensional multivariate gaussian distribution.

   Pass the $Z_e$ sample concatenated with conditions $c$ to the decoder to generate reconstructed input.

   Optimise the network by reducing the overall reconstruction loss with stopping gradient flow to DMM network.

**end for**

**3. SOM Initialisation**

Initialise all the $J$ parent embeddings $Z_q^j$ for SOM clusters from truncated Normal distribution $\mathcal{N}$.

**for** each pretrain-epoch **do**

   For all $Z_e$ encodings, find the closest SOM embeddings $v = argmin_{j \in J} \|Z_e - Z_q^j\|^2$.

   Update the parent embedding $Z_q^v$ and all neighbourhood $Z_q^j \; \forall \; j \in N(v)$, so that embeddings updates towards the assigned cluster.

   Optimize SOM network by minimising the overall SOM loss.

**end for**

**4. Full Training**

**for** each training-epoch **do**

   For each batch, Train DMM, cVAE, SOM and LSTM network End-2-End.

**end for**

**5. Prediction Fine tuning**

**for** each fine tune-epoch **do**

   Model LSTM network to take $\{Z_e^1 \ldots Z_e^t\}$ as inputs and compute $Z_e^{\hat{t}+1}$

   Train Damp model with proportional vectors from $D$ and $L$ to produce damp factor.

   Fine tune the temporal network by minimising the log probability of $Z_e^{t+1}$ with ideal $Z_e^{t+1}$ after multiplying with damp factor.

**end for**

**6. Forecasting Fine tuning**

**for** each fine tune-epoch **do**

   Model LSTM network to forecast $Z_e^{t+1}$ till $l$ steps by applying attention on the past window $w$.

   Reconstruct the $\{Z_e^{\hat{t}+1}, \ldots, Z_e^{\hat{t}+l} \}$ through decoder.

   Minimise the loss of unseen time series data with forecasted data.

**end for**

---

## B.2 TRAINING OF DMM:

Let us assume that we have observed the data for T time steps D = $x_1, x_2, ...., x_T$ where each x $\in R^d$ for N samples (e.g. players) and wish to forecast the next p time steps. At any forecasting step, We divide these T steps in two parts of lengths $C$ and $M$ ($C < M$) referred to as the cluster learning time steps ($C$ steps) and temporal forecasting time steps ($M$ steps) with $C + M = T$.

**Training:** *Step 1: Determine k* We first perform a k-means clustering, choosing data across the $NxT$ data points, with 80:20 train/test proportions. We then fix $k$ using the best silhouette score Rousseeuw (1987). *Step 2: Data Slicing for Training:* In a Markov Model the next state only depends on the prior one. Each state is associated with certain observations and change in the pattern of observations could be related to a change in the state. We refer to a state as "condition" since it associates directly with the term 'condition' in a conditional VAE. As shown in the Figure 2, an observation is represented as proportion list of size k. Each $i^{th}$ entry refers to the proportion of data points that belonged to the $i^{th}$ cluster. Similarly next proportion list will slide post 1 data point, now ending on the $C+1^{th}$ point (red line). MIRAGE adopts a feed forward network with two types of metrics for each sample $i$, to facilitate its learning: 1) error in prediction of the next observation given the immediate past observation as: $\mathcal{L}_{MSE} = \sum_{i=1}^{N} \sum_{t=1}^{T} \frac{1}{d} \sum_{j=1}^{d} \| x_{i,t,j} - \hat{x}_{i,t,j} \|_2$ and 2) a probability distribution of both the predicted and the actual observation via a dense layer with, number of conditions, $s$ as a hyper-parameter using a Categorical Cross Entropy loss; $\mathcal{L}_{Conditional} = - \sum_{i=1}^{N} \sum_{t=1}^{T} \frac{1}{s} \mathcal{P}_{model}(x_{i,t}) \log(\mathcal{P}_{model}(\hat{x}_{i,t}))$, where $\mathcal{P}_{model}()$ is an additional dense layer in the DMM. $X$ is an input set with $N$ time series for $x \in X$. For details on exact steps for training the DMM, refer to AppendixB. The total transition loss of the DMM is given by:

$$\mathcal{L}_{Transition} = \mathcal{L}_{MSE} + \mathcal{L}_{Conditional}$$

## B.3 PLAYER DATASET AND THE METRICS

The player dataset that we acquired consists of about 50,000 unique players' game play feature trajectories. The Table 5 lists the features which are collected per player at a weekly level due to inherent feature sparsity at a day level. The downstream risk model (not a part of this discussion) generates a risk score for every player at a weekly level. This risk score can be directly associated to the player's level of overindulgence. In the above dataset only about **1**% of the data pertains to the players who were sent for the RGP assessment. Hence, this set represents a classic *imbalanced data classification problem.*

*Game Play Risk Model:* The model orders these players possibly on the entire trajectory of their near past risk scores and then sends top few ranked players into a Responsible Game Play (RGP) assessment process which involves taking a self assessment and then followed by a psychological counselling to educate the player on the various do's and don't.We have chosen the time period of observation and forecasting such that these players were sent for RGP assessment at least two weeks *later* from the final date of forecasting. We forecast for 6 weeks ahead. We could do longer but, we were unable to get longer trajectories in the dataset provided.

*Ground Truth for the True Positives:* Though the model performs weekly risk score evaluation and prioritizes certain players for an RGP assessment, there could be more such players who were not identified and hence missed falsely. Unclear about the logic the risk model uses for prioritizing certain players over the others, we experimentally defined a criteria for true positives ground truth creation. We filtered players who have been repeatedly identified with high risk score at least twice before they were actually sent for RGP assessment. With this logic we were able to identify about 75% of the players who were actually sent by the risk model for the RGP assessment. Table 5 lists a few important game play features provided to us.

## B.4 MODELING REAL WORLD DATASETS WITH MIRAGE

Real world MTS datasets like ECL, ETTh, ETTm, etc. are easily modelled by MIRAGE by incorporating a few modifications in the input data to the network, as highlighted in 3.1.1. For each dataset, we performed several experiments to determine the ideal mix of $C$ and $M$ (where $C + M = T$, $T$ =input time steps). For example, for ETTh1 -

| Game Feature | Details |
|---|---|
| cpc | Count of cash games played |
| cat | Count money add transactions |
| acf | Count of add cash fails |
| acf_amount | Amount of cash added for playing |
| fgl | Games lost to won ratio |
| tpd | Total time spent in playing per day in minutes |
| gtc | Number of modes of cash deposit |
| idc | Invalid declarations at the table |
| lng | Count of late night games |
| dld | User set daily add cash limit |
| clh | Count of total times user hit the cash limit |
| clir | Count of limit increase requests |

Table 5: Player Game Features

| Prediction Length | C | M | Average MSE |
|---|---|---|---|
| 24 | 120 | 96 | 0.562 |
| 24 | **144** | **72** | **0.483** |
| 24 | 168 | 48 | 0.692 |
| 48 | 120 | 96 | 0.677 |
| 48 | **144** | **72** | **0.580** |
| 48 | 168 | 48 | 0.772 |

Table 6: Input Data Configurations for ETTh1

## B.5 HYPER-PARAMETER SELECTION AND IMPLEMENTATION DETAILS

Hyperparameters for Mirage were chosen using the heuristics provided in Manduchi et al. (2021) and further tuned using OptunaAkiba et al. (2019) open-source Python library. The following settings of the various weights were indicated to be optimal by Optuna Akiba et al. (2019)

| Hyper-Parameter | Value |
|---|---|
| Alpha ($\alpha$) | 10.0 |
| Beta ($\beta$) | 0.1 |
| Gamma ($\gamma$) | 0.1 |
| Kappa ($\kappa$) | 1.0 |
| Theta ($\theta$) | 0.1 |
| Eta ($\eta$) | 10.0 |
| Tau ($\tau$) | 75 |

Table 7: Final Hyper-Parameters used in Training

| Value of c | ETTh1 | ETTm1 | ECL | WTH |
|---|---|---|---|---|
| 2 | 0.590 | 0.572 | 1.098 | 1.298 |
| 3 | **0.483** | **0.397** | 0.936 | 1.367 |
| 4 | 0.673 | 0.499 | 0.877 | 1.289 |
| 5 | 0.578 | 0.440 | **0.717** | 1.250 |
| 6 | 0.821 | 0.627 | 0.889 | 1.174 |
| 7 | 0.896 | 0.649 | 0.921 | **1.083** |

Table 8: MSE performance on varying $c$ (number of discrete Markov conditions), for a prediction length of 24

### B.5.1 INFRA SETUP AND TRAINING PARAMETER COMPARISONS

For running various experiments on players data as well as other real world datasets, we procured AWS g4dn.xlarge instance for $k = 5$, $s = 3$ and batch size of 128.

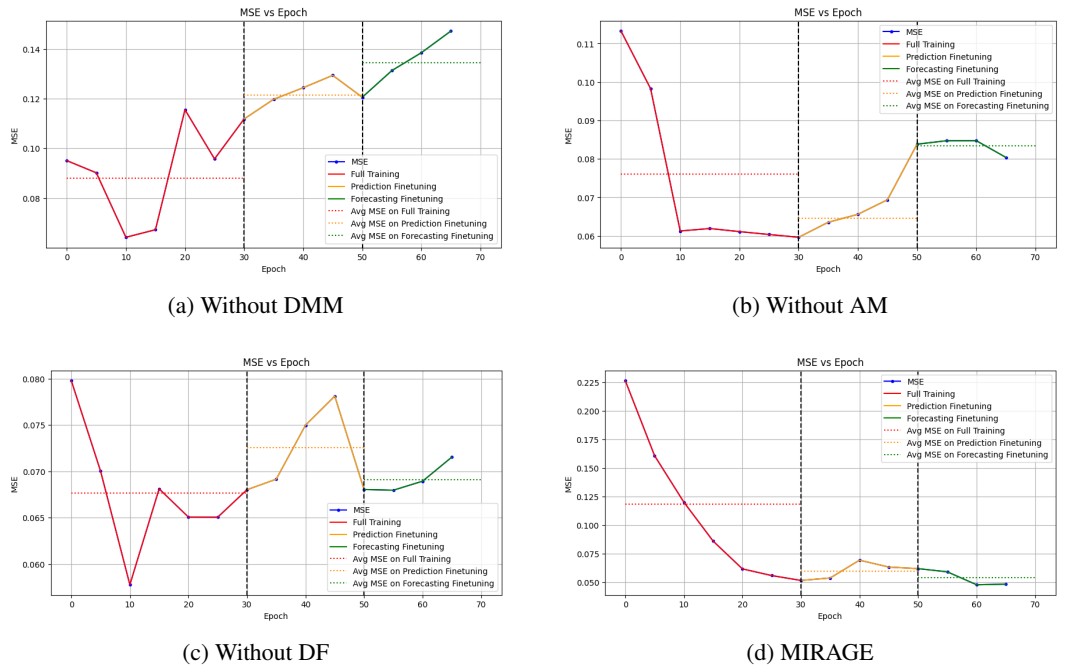

Figure 8: MSE vs Epoch line-graphs on different ablations experiments.

When compared to SOTA transformer models, MIRAGE exhibits a substantial reduction in the number of parameters required for effective forecasting. While transformers are known for their impressive performance across various domains, they tend to be parameter-intensive, often requiring massive computational resources. In contrast, MIRAGE demonstrates a clever combination of Deep Markov Model and Damp factor, allowing it to achieve comparable Mean Squared Error (MSE) results with a significantly smaller parameter count.

### B.5.2 DETAILS OF ABLATION EXPERIMENTS ON MIRAGE

Figure8 shows the variation of MSE in training workflow across all the ablation experiments.

**Incorrect Forecast of player trajectories on Ablation:** In Figure9, we've shown trajectories of 2 sample users, trained - a) Without DMM and DF and b) with MIRAGE and compared their trajectories. With a) trajectories are more confined to their respective neighborhood, due to smoothness factor limiting farther transitions. This could be a problem while modeling non-smooth time series data. With b), we see that the model is able to transition to the farther nodes if there's a sudden change in the data distribution.

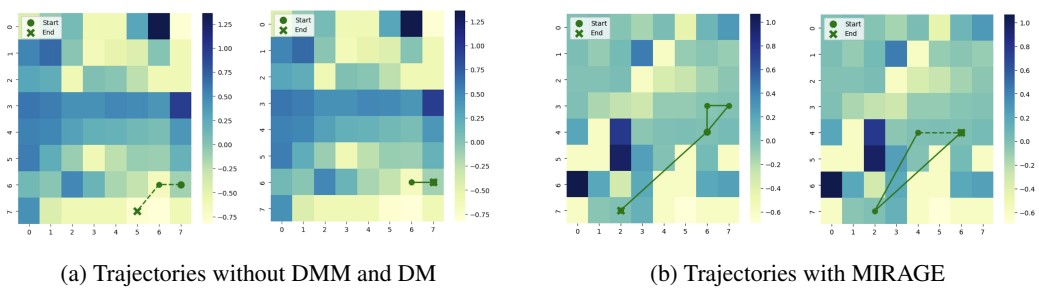

(a) Trajectories without DMM and DM          (b) Trajectories with MIRAGE

Figure 9

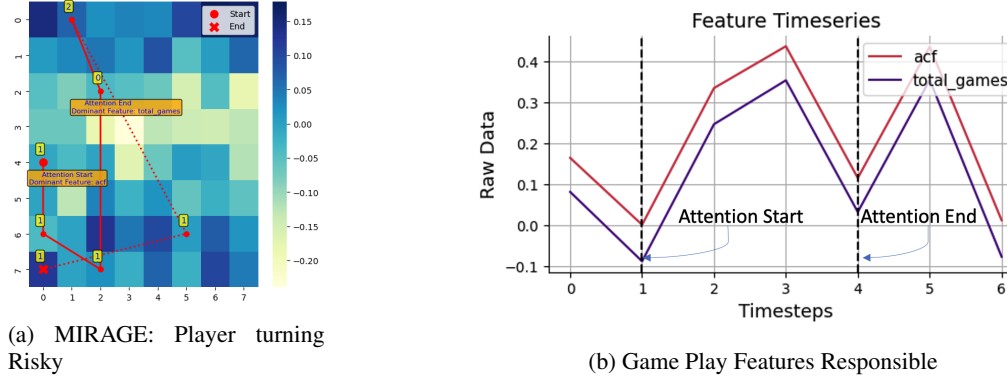

(a) MIRAGE: Player turning Risky

(b) Game Play Features Responsible

Figure 10: (a) Trajectory of a true positive risky player with forecast size=2 (b) Game play data of the responsible feature

# C    EXTRA EXPERIMENTAL RESULTS

## C.1    MORE T-DPSOM TRAJECTORIES ON PLAYER DATA

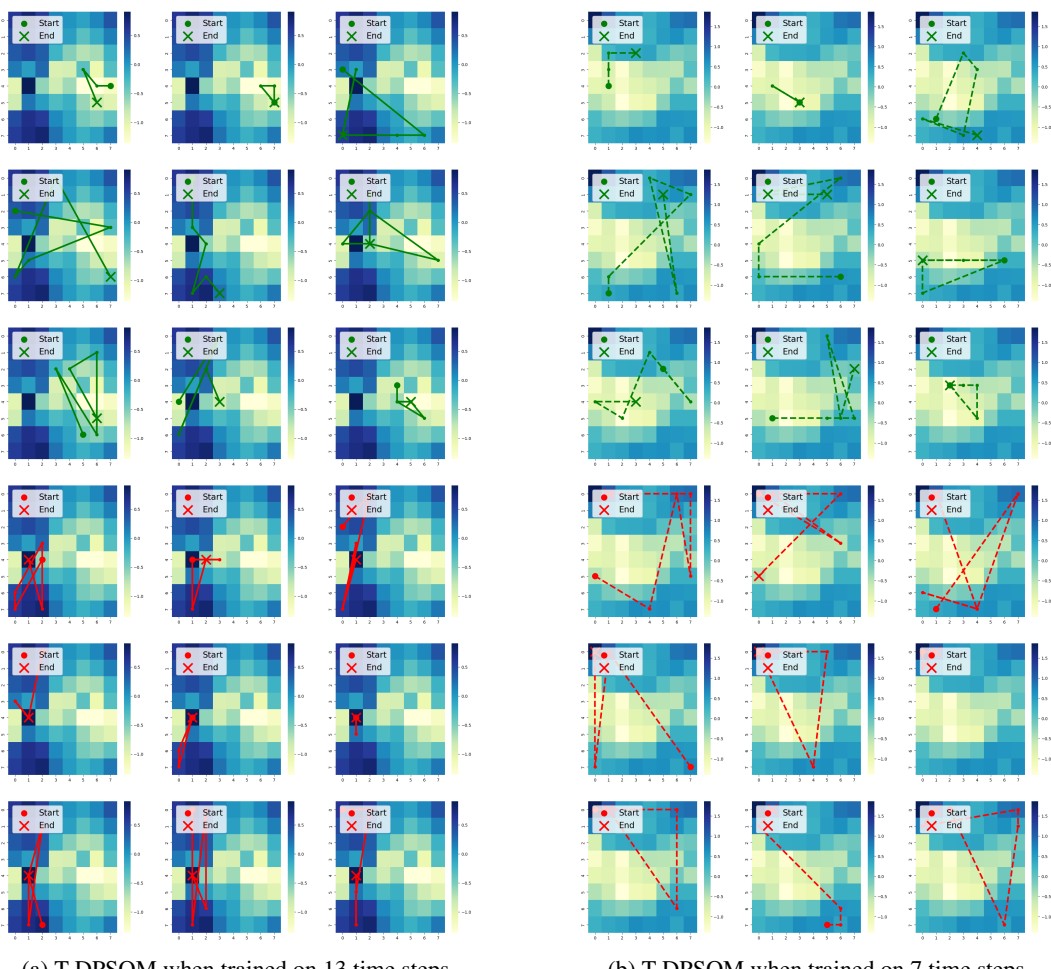

(a) T-DPSOM when trained on 13 time steps

(b) T-DPSOM when trained on 7 time steps

Figure 11: The figure shows how TDPSOM does not do very well with future timesteps on players data

Figure 10 shows the trajectory of another true-positive risky player. The conditions changed from '1' which represents transition of player from dark risky zone to '0' which is less risky and then back to '1'. The attention start and end point are $t_0$ and $t_3$ respectively. This shows that the forecasting him as risky is produced by attending to time-series between $t_0$ and $t_3$ with responsible features as acf and total-games. Similar pattern is observed by plotting input data of these respective dominant features.

Figure 11 shows varied trajectories for the corresponding player when its fully trained (13 weeks) and only interpreted vs. trained for 7 weeks and predicted via LSTM for the next 6 weeks.

- When T-DPSOM is trained on all the 13 time steps of the players data, the overall flow of trajectory appears quite jerky, probably due to non-smooth temporal data and mapping to a single VAE space
- When T-DPSOM is trained on only 7 time steps of the players data, the dark regions doesn't much appear on to the grid to be well captured in a trajectory. Also the trajectory appears more spread out due to again, the non-smooth temporal data which is probably placed too far in the grid.

Summary: the Mapping of encodings to the corresponding embeddings and the objective of smooth transitions perhaps seems not to work out for non-smooth/chaotic data trajectories. This is also evident in the the overall MSE of 0.5 compared to 0.1 in MIRAGE.

*eICU Heatmap with feature importance*: Figure 12 depicts the feature importance derived against the patient APACHE-24 risk score with MIRAGE. Grid shows various features which dominate the respective SOM clusters. This grid could be used as an instant cheat sheet for patient use cases.

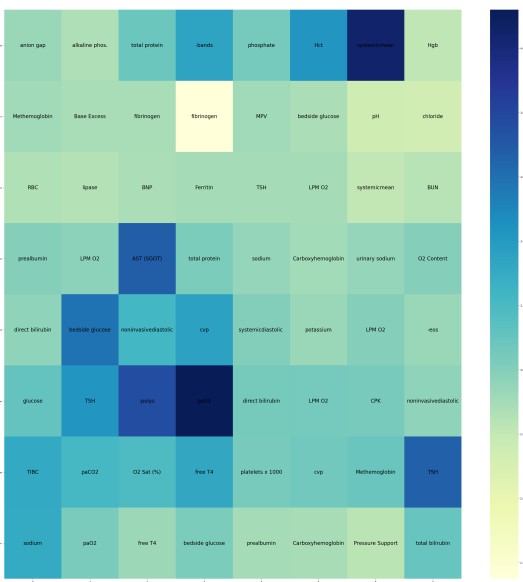

Figure 12: APACHE-24 Risk Score Feature Importance

## C.2  QUANTIFYING INTERPRETABILITY

With player data neighbouring transitions could be non-smooth, hampering interpretability. MIRAGE leverages condition switch as an explanation to justify it. To validate if correlation between a long jump and condition switch is dependable, we created new time series by calculating shortest distance between the clusters assigned to the two successive time steps and correlated with the change in conditions predicted at each point. We found the Pearson's correlation pea (2008) to be primarily high and positive in most cases. Figure 13 shows distribution of the correlation for about 3000 player samples that we forecasted. We take absolute value of the correlation, as both the positive and negative values contribute equally to the interpretations. The distribution is mostly centered

towards a higher correlation coefficient with mean of 0.726 and a median value of 0.797. With mean and the median quite closer distribution seems less skewed to the left. This exercise assures that the transition to farther nodes are mostly associated with a change in the underlying Markov condition and hence is well interpretable.

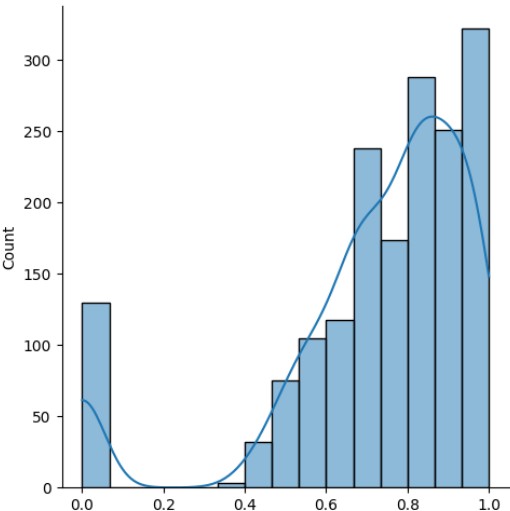

Figure 13: Distribution of Pearson's Correlation values between non-linear trajectory and Markov condition change

### C.3 ADDITIONAL EXPLAINABILITY AND INTERPRETABILITY ON OPEN SOURCE DATASETS

We presented one example of how interpretability helps in understanding forecasts for the various open datasets. through ETTm1 dataset. The Figures 15, 14 and 16 show two interpretable trajectories each. In all of these trajectories, MIRAGE correctly predicts the target variable to be either going from high to low or vice versa. While the bold lines depict the training time steps, the dotted path is the predicted trajectory on the SOM grid. In each of these graphs, the heatmaps correspond to the respective target variables. The scale is divided into 8-10 quantiles depending on the range of the variable, darker zones indicating higher values of the variable and vice versa. The predictions on the corresponding interpretable space move from lighter or less darker to darker clusters (if the value is predicted to rise) and otherwise. The time series plot at adjacent to these heat maps indicate which feature (co-variate) did the target variable attend to and which time step in the already observed time period was the importance attached to. There is no prior available so as to how these various variates influence each other during forecasting. Networks internally modeling the cross dimension dependencies like Zhang & Yan (2023); Lai et al. (2017), though do leverage this knowledge for forecasting, seldomly the networks are able to reveal those details to humans. As a convenient reference Figures 17a ,17c and 17bshows the feature importance for the same corresponding response variable based heatmaps.

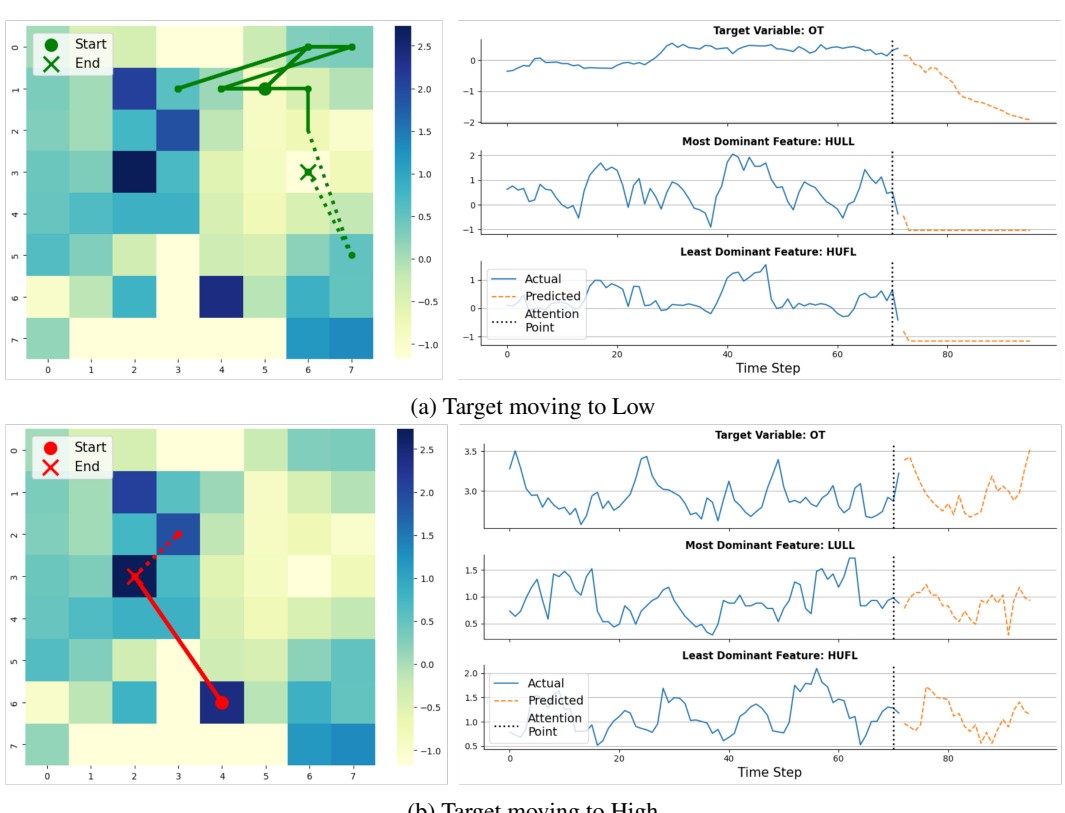

(a) Target moving to Low

(b) Target moving to High

Figure 14: ETTh1:The solid line in the Heatmap denotes input time series and the dotted line denotes the projected time series

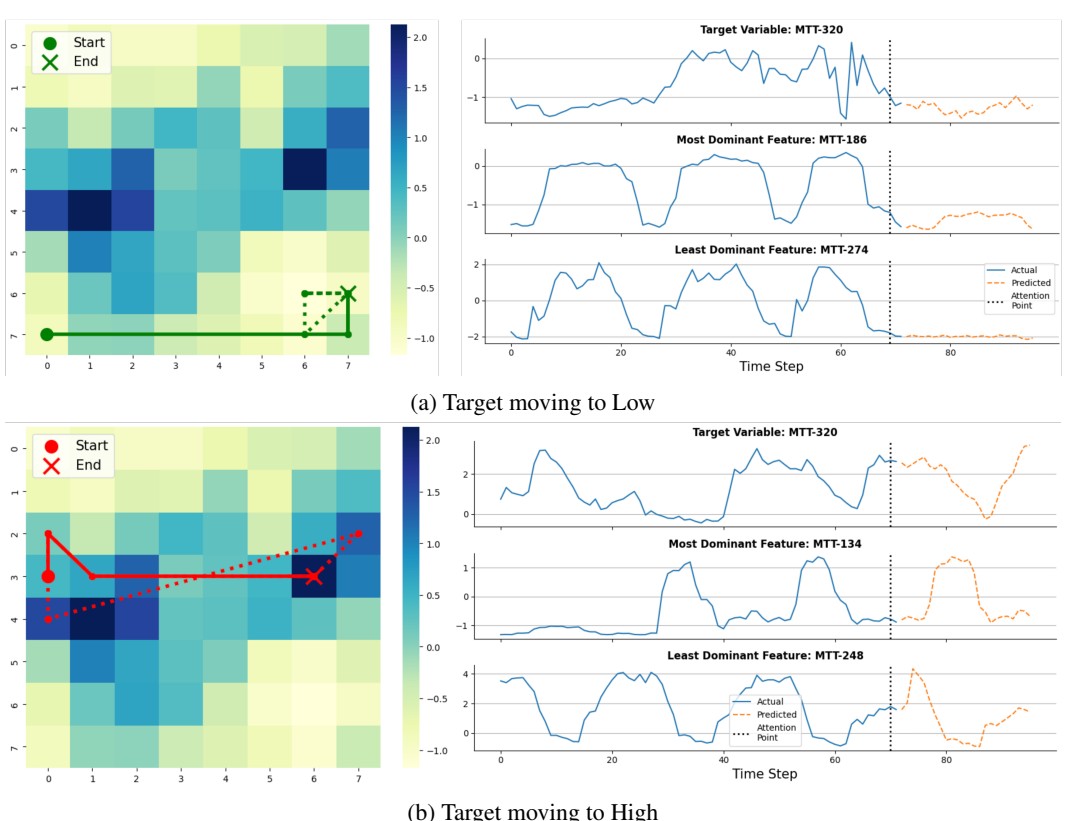

(a) Target moving to Low

(b) Target moving to High

Figure 15: ECL: The solid line in the Heatmap denotes input time series and the dotted line denotes the projected time series.

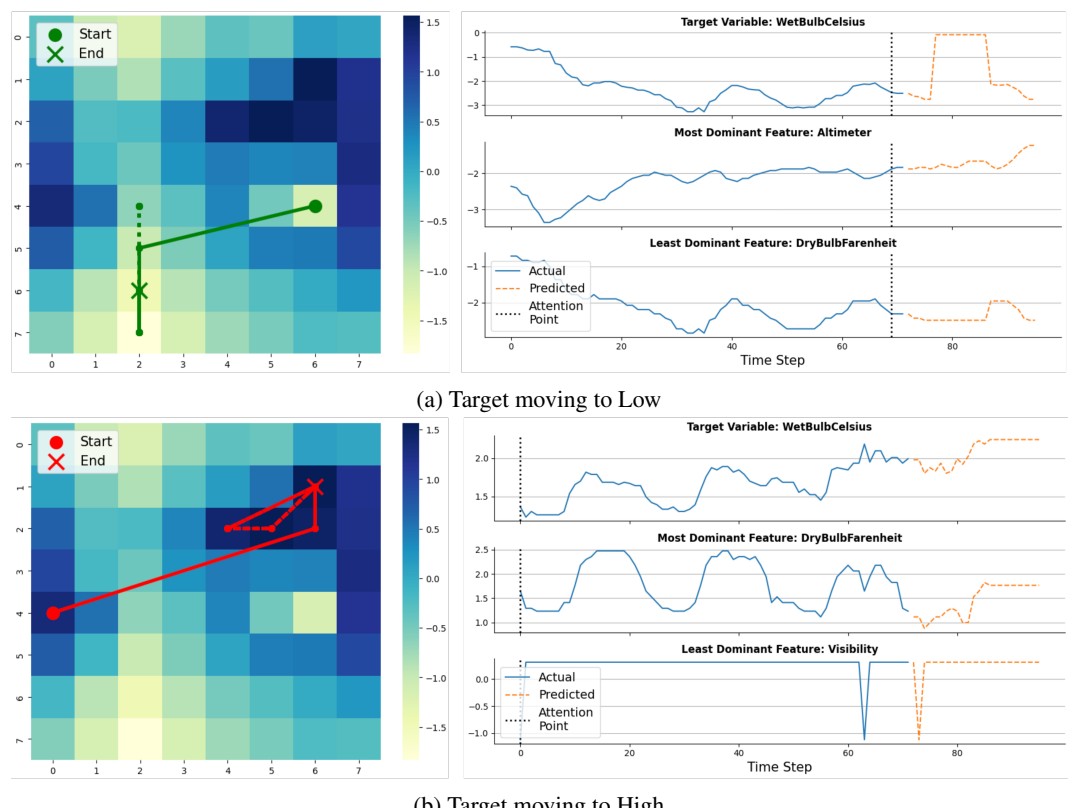

Figure 16: WTH: The solid line in the Heatmap denotes input time series and the dotted line denotes the projected time series.

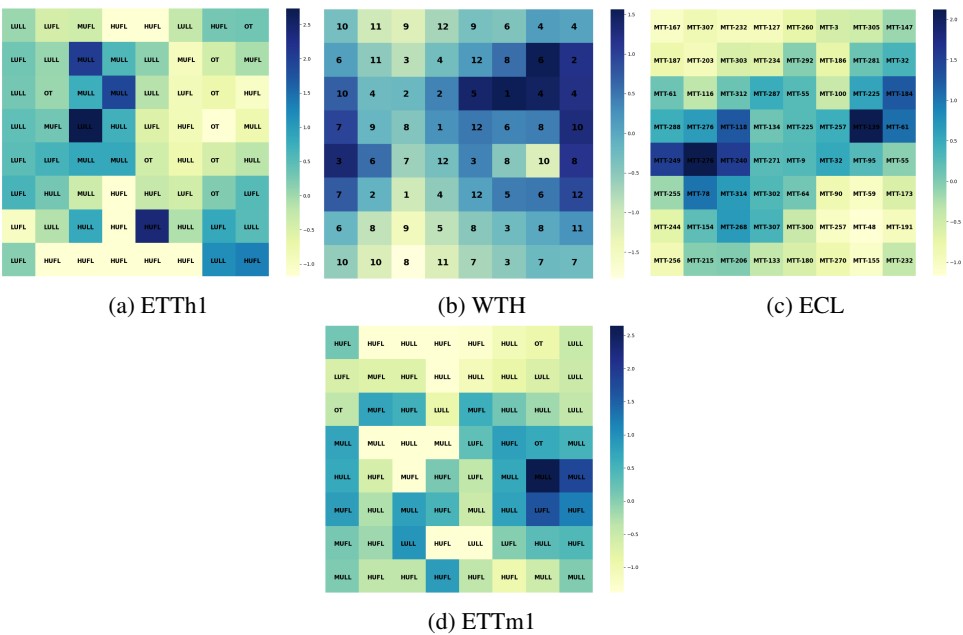

Figure 17: Real World Datasets Feature Heatmaps for the corresponding response variables discussed in the temporal trajectories

