# OpenReview forum: "MIRAGE: Modelling Interpretable Multivariate Time Series Forecasts with Actionable Ground Explanations"
_ICLR.cc/2024/Conference — Submitted to ICLR 2024_

### Official Review · Reviewer_EVwY · 2023-10-30

**Soundness:** 2 fair
**Presentation:** 2 fair
**Contribution:** 2 fair
**Rating:** 3
**Confidence:** 4

**Summary:**

The authors propose a time-series forecasting approach with Deep Markov model (DMM) architecture, an extension over an exsting model T-DPSOM. The DMM module added an extra loss function for state-transition on T-DPSOM which seems to perform well over T-DPSOM for fine-tune forecasting model.

**Strengths:**

Used analysis on multiple public and real-world data like game player and medical ICU data.

**Weaknesses:**

1. What is the exact novelty of Mirage over T-DPSOM paper? What is the motivation behind DMM architeture?
	- Table 1, Mirage underforms crossformer over MSE/MAE score for all public data.
	- Table 2, performance only shown for T-DPSOM for ICU and Mirage analysis only shown for player data, where mirag has negligible improvement over crossformer.
	- Which dataset is used for Table 3 performance? Is it average performance on all data?

2. Table 1, authors bolded the values of Mirage in MSE, where clearly Cross-former is the lowest scores. Is this mistakes been done to create misinterpretation fir the reviewers or just type error?

3. Overall the paper is very hard to conceive, specially Sec 4 Evaluation.
	- For someone not in medical/health data expert, the real world medical data needs a bit more explanation. What do the timeseries represent, length of the sequence, condition, etc.? Is biluribin level a time-series feature for patient?
	- I am not clear on the part how the trajectory steps are being tracked and how the heatmap being generated? Is the heatmap represent the attention weights?
	- used many abbreviations w/o terminology, e.g., acf
	- Table and Fig. captions are not self-explanatory.

4. I had to do back-n-forth multiple times to understand the entire loss function. Using proper eq. labels would have helped a lot.

**Questions:**

1. What is L_{DPSOM}? Could not find the equation. Without reading the T-DPSOM paper, this will be hard to understand the Mirage paper. If there is an short background of T-DPSOM model, its pros and cons, where it performs and why it fails were explained in the beginning would have been easier.

2. Evaluation analysis explanations are hard to conceive. Authors first start with Fig. 5 then suddenly jumped back to Fig. 3

3. Reviewer is not clear on how the figs (3-5) are generated.

4. Overall, the Mirage results showing underperfroming crossformer on both public and real-world data. Then what is the usefulness of Mirage architecture?

---

> ### Author Response · Authors · 2023-11-21
> **Response to Reviewer 3**
>
> Dear Reviewer,
>
> Thank you for your valuable feedback which has helped us reshape our paper.  We would like to firstly address your primary comment on what is the exact novelty of MIRAGE and purpose of  DMM.
> Transformers based Multi-variate Time Series (MTS) forecasting has made significant progress (with very negligible errors). However, with the advances in AI architectures, model interpretability and explainability have not received much attention (partially also because these terms have not yet been fully understood). Also, existing knowledge has well established exclusivity between these three objectives - prediction accuracy, interpretability and explainability and often the prior objective is more sought after.  However, in critical situations like predicting a death in an ICU or sudden gaming overindulgence affecting ones mental well-being; an accurate prediction (forecast) without a contributing evidence (explanation) is irrelevant.
>  In our paper, we refer to the only available SOTA network focussed on interpretability - SOM-VAE (ICLR 2019) and its refined variant T-DPSOM. But here, authors make an assumption on data linearity over time. We quantify that our non-linear, a-periodic time series data of players’ gaming indulgence violates these constraints and warrants a relook at the solution( Section Motivation).  MIRAGE comes up with a novel formulation based on the fundamental intuition that the latent state of a process helps infer if the immediate next data point is going to be linear or non-linear.  MIRAGE incorporates a gradient based method of learning a DMM ( Discrete Markov Model ) as a part of the T-DPSOM model.  However, yet it becomes hard to guarantee a smooth temporal trajectory and how MIRAGE trains a damping factor from the DMM output to supplement the interpretability is MIRAGE. We now provide additional quantitative metrics to validate MIRAGE driven interpretability for a non-linear player dataset. We use interpretability to also identify (classify) future player trajectories as overindulgent or healthy. We report improved accuracy (precision/recall) with MIRAGE vs the T-DPSOM model in the Evaluation Section
>
> - Comment regarding under-performance of MIRAGE w.r.t. Crossformer
> Paper states that MIRAGE MSE results do not compete with any of the Transformer based models, which are the SOTA models for predictions. MIRAGE does better than LSTNet (Lai et al. , 2017) (which is one of the best industry standard benchmarks after the Transformer based models) on ETTh1 and ETTm1 datasets (quoted in Crossformer - ICLR 2023). LSTNet specifically focuses on MTS forecasting and uses a CNN to extract cross-dimension dependencies and LSTM for cross-time dependencies. We have corrected our bold markings of figures which were intended to highlight against LSTnet and had mentioned so, in the writeup.  Our humble apologies the confusion.
>
> - For comments related to lack of clarity etc.
> We have now updated our pdf handling all the mentioned labelling, missing data, lack of clarity, and figure overcrowding.  We have re-arranged our writing to clearly explain our motivation and the differentiation we add. We have refined our description of the model and architecture adding much more clarity and re-discussed some of the evaluation findings.
>
> - For Medical Data
> eICU data is a time series data for patients' vital signs  (blood and other body parameters) out of ICU, from the eICU Collaborative Research Database ( Goldberger et al.). We have added more details on how the heat maps and graphs were generated in the new revision of our paper. To summarise, colour of a cell in the heat map (2-D SOM map of the centroids) represents the mean of the metric under evaluation ( -e.g Apache health score or Players overindulgence score) of the points which map to it. Each cell represents SOM cluster and scores were normalised [0-1] before taking mean.
>
> - Overall MIRAGE MSE results outperform the only available SOTA benchmarks for interpretability - SOM-VAE and T-DPSOM and LSTnet which is a pure time series forecasting model. MIRAGE underperforms to Crossformer and its variants, which however do not address interpretability and explainability
> - MIRAGE shows impressive explanations for player overindulgence and  ICU re-admissions or healthy discharge cases.
> - MIRAGE extends SOTA networks for interpretability on non-linear datasets and quantifies the results (newly added).
> - For the open real world datasets (ETTh and others) we contend that MIRAGE is probably ahead in time, to reveal interesting facts on which covariates in the data gives an early indication of rise in electricity demand in a city with no ground truth available to validate.
> - Lastly, we believe that open sourcing MIRAGE will significantly elevate the state of preventive health care with proactive causal and actionable explanations.
>
> We would be greatly obliged if you could give one more read to our updated paper version.
>
> Thank you.

---

### Official Review · Reviewer_QroT · 2023-10-30

**Soundness:** 1 poor
**Presentation:** 1 poor
**Contribution:** 2 fair
**Rating:** 3
**Confidence:** 4

**Summary:**

The goal of this work is to provide an interpretable model for time-series forecasting. The proposed method involves a clustering stage to determine a general Markov state structure using the first part of a trajectory, with learned transition dynamics. The second part of the trajectory is used for prediction/forecasting after being mapped to an interpretable SOM-VAE latent space (proposed in prior work). Attention weights and SHAP values are extracted on top of the proposed model to provide explainability.

**Strengths:**

* Interpretability + time-series modeling remain a important, open problem in the literature.
* Some interesting ideas to challenge modeling assumptions in prior work (e.g. smooth changes in latent space).
* A variety of time-series datasets are considered.

**Weaknesses:**

* The proposed method reads as a large, complex collection of unmotivated components, and little insight is given as to why they are necessary. I would encourage authors to narrow down the key, novel elements of their method and to propose a more fundamental motivation and rigorous analysis of their added value.
* Empirical results consist of some illustrative examples (are these random examples or cherry-picked?) and few rigorous numerical analyses.
* Not sure I understand the premise of interpretability/explainability used in paragraph 1 of the introduction.

* Presentation: I found the paper confusing and tiresome to read.
  * Figure 1 is overcrowded and confusing. Most elements are undefined. Poor quality (delineation of underlying elements, poor alignment).  Figure 2: what is CCE loss? What is shift loss?
  * Figures in the experimental results section are generally illegible with little or no labeling. Fig 4, for example: what are alternate features? What do the medical variables correspond to, and how does this correlate to medical insight? There is no legend for what dashed/solid lines correspond to.
  * A lot of notation is undefined. e.g. what is index $w$? Difference between $Z$ and $z$?
  * Definition of abbreviations (many in abstract!)
  * Language is overcomplicated (see abstract again) with many undefined or unclear ideas: “proactive comprehension of trajectory to an extremity”, “observations are competitively mapped”, “part of its learning stride”, “collaboratively trained”, “results are assuring”, “Recollect that”, “agrressive” typo, “movement to criticalities on temporally chaotic datasets”, “Providing Ground explanations” etc.
  * Please put references in parentheses when they  do not form part of the sentence.
  * Missing hyphenation between words (“outcome oriented”, “down stream”, “scale varying”) and punctuation.

Unfortunately, with such major issues unaddressed, the manuscript is not ready for publication.

**Questions:**

* P1, “The psychological imprints…” how does this example illustrate lack of smoothness? I agree with the last sentence ("the factors affecting the future predictions (co-variates) are not completely observed, measurable, or generalizable") but don't see how this relates to "non-smoothness".
* what does "scale-varying/variably scaled features" mean? Isn’t this inherent to any TS data? If variable scale is an issue, why not just normalize? And how does MIRAGE specifically tackle this?
* Why use an LSTM and not a transformer as prediction architecture?
* How is $C$ determined?
* Why is the MSE of MIRAGE on eICU data (Table 2) not reported? Perhaps authors could report reconstruction error in addition to  forecasting performance.
* How do authors determine that “interpretations appear quite smooth” in Fig 7?
* Could authors provide numerical results that support the interpretability/correctness of latent trajectories, beyond the few qualitative examples proposed?
* Also would be curious to understand how HUFL can be interpreted as contributing to a drop in variable OT, whereas the least dominant feature LULL also shows a trend over a similar timescale…

---

> ### Author Response · Authors · 2023-11-21
> **Response to Reviewer 2**
>
> Dear Reviewer,
>
> Thank you for your valuable feedback which has helped us reshape our paper.  We have now updated our pdf handling all the mentioned labelling, missing data, lack of clarity, undefined notations and figure overcrowding, hyphenation and parentheses issues, to the best of our ability.
>
> We have rearranged the writing to clearly explain our challenges/motivation and the differentiation we add (Section Background and Motivation). We also discuss  interpretability vs. explainability and how they differ. We have refined our description of the model and architecture and re-discussed the evaluation findings.  Apologies for the confusion with the terminologies. We have much simplified our writing now.
>
> With regards to your specific questions:
> - We have corrected the paper w.r.t first 2 points (Section Background and Motivation)
> - Why not Transformer instead of LSTM :  Agreed, very appropriate suggestion, especially to the prediction variability question we pose in the conclusion section of the paper. Transformers have proven to provide the best results so far, with negligible errors. In our comparison studies we do mention that if prediction accuracy is the end goal, Transformers based extensions (Crossformer or Scaleformer - ICLR 2023) is the way to go.  Exclusivity of training objectives between prediction accuracy, interpretability and explainability has led the researchers to choose only one objective at a time.  Transformers, due to complex mapping from features to intermediate representations, to the best of our understanding there is not much exposure on explanations or interpretability so far, except for the Temporal Fusion Transformer (more details in response to Reviewer 1).
>
> MIRAGE, attempts all three objectives - prediction, interpretability and explainability, at once. It trains an attention layer, to expose explainability with a definite time reference for a proactive action. This time window cannot be generalised (unlike the Temporal Fusion Transformers) but has to be very specific to a user, e.g a particular player who is going to be overindulgent or a patient going for ICU readmission in the coming future.  So, replacing LSTM by a Transformer would need a detailed study on preserving the explainability across its intricate mappings from feature to the latent space.
>
> - How is c derived - c is the total number of discrete Markov states DMM could predict. They are derived empirically through experimentation. We leveraged Optuna framework for this.  We have now presented empirical results on varying c on open datasets in the Appendix (Section C, Table 8).
>
> - MSE on eICU data for MIRAGE: Apologies, MSE was reported in the wrong column. We did not intend to report MSE on Crossformer as we already see it's much better. Also, as our focus is on interpretable predictions. Comparisons on MSE with Crossformer and LSTnet are reported for the open real world datasets as these are the standard benchmarks for those.
>
> - We have improved our explanation on what interpretability and smooth interpretations mean in the Section - Background and Motivation
>
>  - Interpretability: We know, interpretability is vaguely defined in the AI systems as of today. In this paper we further extend the first of its kind established notion of interpretability in SOM-VAE (ICLR 2019) paper. Unfortunately, to the best of our knowledge there no standard metrics for its rigorous evaluation. SOM-VAE primarily, establishes its claim by visual trajectories on eICU data. We have published alike visual trajectories for all types of datasets and more in the Appendix.  However, based on your feedback, we now report a metric based on Pearson's correlation to represent correlation between a linearity of MIRAGE trajectory for non-linear temporal progression and the corresponding transition in the discrete state of the process which generated it (Section Evaluation). Its quantification (numerical results) on the overall player data is also added in the Appendix C.2.  We use interpretability to also identify (classify) future player trajectories as overindulgent or healthy. We report improved accuracy (precision/recall) with MIRAGE vs  the T-DPSOM model in the Evaluation Section
>
> - HUFL : Addressed in our writing.
>
> Explainability - Explainability highlights features in the input space that caused a particular trajectory and hence rigorous analysis was not possible and had to be presented case to case basis. We provide interesting examples on each of the available datasets. Patients' health data (eICU), due to high levels of privacy, lacked any ground truth on what caused deaths for certain patients.  MIRAGE revealed intuitive explanations (drop in bilirubin for a liver diseased patient, indicated recovery). We have few more examples for each dataset in the Appendix. We have published our code, results as well as datasets.
>
> We would be greatly obliged if you could give one more read to our updated paper version.
> Thank you.

---

> > ### Comment · Reviewer_QroT · 2023-11-23
> >
> > Thank you very much for your detailed response.
> >
> > I believe many proposed changes improve the quality of the paper. It would have been helpful to highlight them so we don't have to go through the whole paper again. Figure 2 is much better than I remember it, but could maybe benefit from even further simplification for the reader.
> >
> > Still, while I believe the method proposed is interesting and the experimental results promising, I retain the following criticism and therefore retain my score:
> > * Section 1 has become very challenging to read and understand. It is still unclear to me what Table 1 shows (what is in bold, what do the tests measure?) and how it motivates MIRAGE to the unfamiliar reader.
> > * Figures 5,6,7 are still illegible and many presentation issues remain. Please refer to my comments above and carefully proofread.
> > * I believe your results in quantifying interpretability are key to supporting your claims and to bringing this work above the acceptance threshold. Although I found it challenging to understand (presentation must be improved!), I think I see what the authors propose, i.e. measuring the correlation between a change in modelled (VAE) condition and a change in the underlying time-series (discretised into a Markov chain via clustering). I find this promising, but this only focuses on the 'non-smooth' transitions: how could you quantify the interpretability of smooth predictions and of dominant features? In my opinion, this is as important as illustrative examples to demonstrate interpretability. A possible solution could be to ask humans to blindly rate the interpretability of your model output, of T-DPSOM and maybe of a simple time-series model with post-hoc SHAP analysis.

---

> > > ### Author Response · Authors · 2023-11-23
> > > **Response to Reviewer2**
> > >
> > > Dear Reviewer,
> > >
> > > - In table 1 we try to quantitatively assess non-linearity. From our understanding of the past literature, we were aware of the Runs Test (reference to the paper provided) which is quite an old statistical technique to determine whether a sequence of data within a given distribution have been derived with a random process or not with NULL statistic being that the values are random. We see that p-value is quite significant leading to the acceptance of the NULL statistic.  The second metric measures 1-lag correlation or in another sense difference between the first order difference between the consecutive values, and is also a quite old method. The eICU dataset using by SOM-VE and T-DPSOM based models shows very high correlation (~95%), whereas our custom dataset reports a much lower value.  We can definitely improve our description of the same in the paper.
> > >
> > > - With regards to Figures 5,6 and 7, definitely could be further improvised. We hope that you understood the basic premise in the context of what dominant attentive features mean and how interpretability is defined and measured by us.
> > >
> > > - We perhaps missed to mention that MIRAGE handles both linear (like eICU and ETTh/m) as well as non-smooth (player dataset) time series. The DMM (Discrete Markov Model) is capable of predicting if the next transition is coming from the same condition or a different one. For instance in Figure 7 (ETTm dataset) the heat map trajectory generated by MIRAGE is quite smooth. By smooth, we mean that in the consecutive time step the SOM centroid either remains the same or moves to the neighbouring centroid (SOM centroid in the left, right, up or down) which is not the case for the player (non-linear) data (Fig 6) which shows long jumps. But both these trajectories have been generated by MIRAGE.  We have also performed active validation on trajectory based prediction for about 3K players and we have reported much improved precision/recall numbers in the Evaluation Section over the state of the art. Definitely, your suggestion on human based assessment is valid and we will consider.

---

### Official Review · Reviewer_nYfL · 2023-11-08

**Soundness:** 2 fair
**Presentation:** 2 fair
**Contribution:** 2 fair
**Rating:** 5
**Confidence:** 2

**Summary:**

The authors introduce a Multi-variate Time Series (MTS) forecasting model designed to address non-smooth data and deliver high-quality interpretable forecasts. The model, named MIRAGE, comprises multiple components, including a Deep Markov Model (DMM) for handling non-smooth data, an Attention Module (AM), a Damping Factor (DF) element, Forecasting Fine-tuning (FFT) element, and a Self Organizing Map (SOM). The DMM manages non-smooth data, while the AM, SHAP (SHapley Additive exPlanations) analysis, and the SOM contribute to model interpretability.

**Strengths:**

I value the paper for its specific insights into how the MIRAGE architecture can be extended, offering a clear path for further research and development. Additionally, the paper's overarching emphasis on addressing the interpretability challenge in the realm of time-series data is commendable and contributes to a better understanding of complex forecasting models.

**Weaknesses:**

1. The analysis for the other datasets is somewhat limited. Since the paper primarily revolves around model interpretability, a more in-depth examination of how the features in these datasets are employed in making predictions would enhance the comprehensiveness of the research.
2. There are labeling issues in Table 6/1.
3. Figure 2 isn’t labeled as such.
4. The text within the figures should be presented in a larger font size to improve readability, ensuring that readers can easily interpret the visual content.
5. The description of the MIRAGE model is difficult to follow, which may pose a barrier to understanding its functionality.

**Questions:**

Have you considered comparing your model with architectures based on the Temporal Fusion Transformer?

---

> ### Author Response · Authors · 2023-11-21
> **Response to Reviewer 1**
>
> Dear Reviewer,
> Thank you for your valuable feedback which has helped us to reshape our paper.  We have now updated our pdf handling all the mentioned labeling, missing data, lack of clarity issues.  We have rearranged our writing (pls refer to the new paper pdf) to clearly explain our motivation and the differentiation we add. We have refined our model description and architecture and re-discussed some of the evaluation findings with few additions.
>
> Exclusivity between prediction accuracy, explainability and interpretability has led to most of the SOTA models target a single objective, prediction accuracy being the most common.  SOTA models available on Interpretability assumes time series with linear temporal progression (SOM_VAE - ICLR 2019). Our non-linear , a-periodic time series data of players’ gaming indulgence violates these constraints and needs a fundamental relook.
>
> With regards to your concern on the in-depth analysis -
> Forecasting Accuracy - We present MSE results on prediction accuracy for our own player dataset, as well as eICU and few other real-world datasets and compare them with SOTA benchmarks. We have open sourced our tool, the player dataset as well our notebooks.
>
> Interpretability - As we know, interpretability has been vaguely defined in the AI systems as of today. In this paper we further extend the first of its kind established notion of interpretability in the SOM-VAE paper. Unfortunately, to the best of our knowledge there are no standard metrics for its rigorous evaluation. SOM-VAE primarily establishes its claim by visual trajectories on eICU data.  We present similar visual trajectories. However, based on your feedback we report a metric based on Pearson's correlation to validate interpretability (Section Evaluation). General quantification on the overall player data is also added in the Appendix C.2. We use interpretability to identify (classify) future player trajectories as overindulgent or healthy. We report improved accuracy (precision/recall) with MIRAGE vs the T-DPSOM model.
>
> Explainability - Explainability highlights features in the input space that caused a particular trajectory.  To our understanding providing a comprehensive validation is difficult unless the insights are at a generic level. In eICU and the player use case, each trajectory is unique and hence generic explanations would not be of much value.  We provide interesting examples with attentive features on each of these datasets on how MIRAGE is able to pinpoint the exact causal feature (especially for eventualities), most of which have been validated either with the available ground truth or with a rigorous manual evaluation.
> There are no studies yet on the features and their dependencies/influence on the response variable for the real world datasets (like ETTh). One may say that for these datasets MIRAGE is probably ahead in time or rather by using our model, the research community can start exploring much deeper explanations. For instance, how can one prevent the electricity demand from rising by probably controlling one or more  attentive covariate features? Patients' health data (eICU), due to high levels of privacy, lacked any ground truth on what caused deaths for certain patients.  MIRAGE revealed intuitive explanations (drop in bilirubin for a liver diseased patient, indicated recovery). We believe that open sourcing MIRAGE will significantly elevate the state of predictive and preventive health care with proactively actionable explanations. Additionally, we provide feature importance maps for all datasets, which can be used as a guide to associate a trajectory with the corresponding attentive features (Appendix C.1 and C.3 ). More explainability examples have also been provided in the Appendix C.
>
> Temporal Fusion Transformer:  We found this work quite promising and relevant as it tried to provide explanations for forecasts. Few differentiating factors between TFT and MIRAGE to best of our understanding:
> TFT have used interpretability and explainability interchangeably but we’ve used interpretability to answer the “why” of the prediction output by visualising the forecast on SOM map at each interval and used explainability to answer the “how” of the prediction output by obtaining the timestamp and associated feature responsible for the future prediction.
> TFT paper has not published MSE metric to compare with the latest SOTA transformers models in MTS forecasting like Crossformer, Scaleformer (ICLR'23) etc..
> TFT primarily discusses the extraction of overarching trends and patterns from the data, rather than delving into the specific causality of predictions on a case-by-case basis. For instance, in our study, we demonstrated how the model attends to changes in Hct levels (eICU) to predict the likelihood of a patient not surviving. We have referenced TFT work in our related work section.
>
> We would be greatly obliged if you could give one more read to our updated paper version.
>
> Thank you.

---

### Meta-Review · Area_Chair_8WZF · 2023-12-06

**Metareview:**

The paper proposes a novel model called MIRAGE for interpretable time-series forecasting. While MIRAGE holds significant promise for addressing the long-standing challenge of interpretability in this domain, its current presentation suffers from several critical shortcomings that limit its effectiveness and hinder its full potential.

On the positive side, MIRAGE prioritizes providing explanations for its predictions, a crucial aspect often lacking in traditional time-series models. Additionally, the paper presents a clear and extensible architecture, outlining MIRAGE's inner workings and offering a roadmap for future development. Furthermore, the model is evaluated on a diverse set of public and real-world datasets, demonstrating its potential for broader applications.

However, MIRAGE currently suffers from a lack of clarity. Confusing figures, undefined notation, and unclear explanations make it difficult for readers to grasp the model's functionality and understand the underlying principles behind its design. Additionally, the paper fails to adequately motivate the inclusion of key components like the Deep Markov Model (DMM), leaving the reader questioning their necessity and contribution.

Furthermore, the evaluation of MIRAGE lacks rigor and consistency. The paper presents comparisons with other models that are often inconsistent and difficult to interpret, making it challenging to assess MIRAGE's true performance and identify its strengths and weaknesses.

Some of these concerns have been partially addressed with the reviews, but the paper cannot be accepted in its current form. By addressing these critical flaws and focusing on clarity, motivation, and rigorous evaluation, MIRAGE has the potential to evolve into a truly effective and interpretable time-series forecasting model.

**Justification For Why Not Higher Score:**

- Clarity: The paper is difficult to read due to confusing figures, undefined notation, and unclear explanations.
- Motivation: The novelty and motivation behind key components are not clearly explained.
- Evaluation: The paper lacks rigorous analysis and interpretation of results.

**Justification For Why Not Lower Score:**

- Interpretability: The paper emphasizes interpretability, a key challenge in time-series forecasting.
- Extensibility: The MIRAGE architecture is clearly presented with a roadmap for future research.
- Experiments: The model is evaluated on diverse public and real-world datasets.

---

### Decision · Program_Chairs · 2024-01-16

Reject